# Online Boosting-Based Target Identification among Similar Appearance for Person-Following Robots

**DOI:** 10.3390/s22218422

**Published:** 2022-11-02

**Authors:** Redhwan Algabri, Mun-Taek Choi

**Affiliations:** 1Research Institute of Engineering and Technology, Hanyang University, Ansan 15588, Korea; 2School of Mechanical Engineering, Sungkyunkwan University, Suwon 16419, Korea

**Keywords:** person identification, multiple features, online boosting, mobile robot, person following

## Abstract

It is challenging for a mobile robot to follow a specific target person in a dynamic environment, comprising people wearing similar-colored clothes and having the same or similar height. This study describes a novel framework for a person identification model that identifies a target person by merging multiple features into a single joint feature online. The proposed framework exploits the deep learning output to extract four features for tracking the target person without prior knowledge making it generalizable and more robust. A modified intersection over union between the current frame and the last frame is proposed as a feature to distinguish people, in addition to color, height, and location. To improve the performance of target identification in a dynamic environment, an online boosting method was adapted by continuously updating the features in every frame. Through extensive real-life experiments, the effectiveness of the proposed method was demonstrated by showing experimental results that it outperformed the previous methods.

## 1. Introduction

Robots have the potential to be used in several practical applications, and they will be employed to assist people in performing their daily activities in the next decade [1,2,3], such as carrying heavy objects, assisting the elderly, assisting medical staffs in rehabilitation, guarding, and guiding. With the growing number of human detection techniques [4,5] and control systems [6,7,8,9] in various working environments, the abilities of such systems to recognize, track, and identify objects have become increasingly important. In particular, the environment in that humans works side by side with robots. Recent rapid advancements in artificial intelligence techniques and the capability of robotics technologies have resulted in enhanced comprehension of artificial systems comparable to human-like performance in specific applications.

Robust person-following in realistic environments is one of the most critical functions of a mobile robot. Here, the main challenge is to follow a specific target person in a dynamic environment, comprising people wearing similar-colored clothes and having the same or similar height. However, the system must address such a challenge within the available sensors’ possibilities and mounted on the robots.

Person identification based on tracking is often used by robots to follow a target person; this refers to identifying a target person over time using the person’s characteristics and localization. Many studies on person-following techniques have been published. One of the earliest approaches for person-following robots used computer vision to track people based on appearance [10]. Some of these studies employed sensors, such as stereo cameras [11] and laser scanners [12]. Compared to cameras, laser scanners provide poor information about human features. However, affordable sensors can identify people based on their height and appearance. Researchers have employed red, green, blue, and depth (RGB-D) cameras in recent applications, such as the Orbbec Astra [13] and Kinect [14]. These cameras provide synchronized color and depth data and are highly suitable for indoor environments. Moreover, they have acceptable measurement characteristics and are affordable and readily available. Therefore, building such robotic systems is easy.

This study introduces a new person identification framework for mobile robots that classifies people based on online boosting by merging many features into a single feature. In this framework, the robot first detects and tracks people using a deep neural network technique that receives two-dimensional (2D) image sequences from an RGB-D camera mounted on it. This approach allows us to exploit the deep learning technique to extract features of people and then input these features into the online boosting model to classify people as target or non-target persons. Using an online boosting model, the proposed framework re-identifies the target person if the robot loses tracking owing to occlusion or some other event, which is based on information learned before losing track.

The main contributions of this study are summarized as follows: First, we present a novel vision-based person-identifying approach using four features. This approach extracts a clothing color feature from the upper body, estimates the height relative to the ground plane level and location relative to the center of the images, and calculates the IoU (intersection over union) between the current and last frames using an RGB-D camera. The novelty of our approach is that it obtains more features of the target person in terms of the height difference, localization difference, and IoU data between the target person and other people in the current frame and the last tracked frame to increase the robustness of classification. Second, we comprehensively evaluate several online boosting algorithms and weak learners in terms of accuracy and speed to integrate these features into a single feature based on the best online boosting model and weak learners. Third, we designed a system that can be generalized and applied to any target person without prior knowledge. This is performed by using the mouse to extract features when choosing the target person and transferring each feature into a normalization case. Fourth, the person identification framework was implemented on an actual robot and verified in a realistic indoor scenario through intensive experiments using the four proposed features. Other experiments were conducted using only the two features adopted in [3] for comparison based on the features used.

The remainder of this paper is organized as follows. Section 2 provides an overview of the related work. The proposed human-following robot methodology is described in Section 3. The empirical results and discussion are presented in Section 4, which is followed by conclusions and future work in Section 5.

## 2. Related Work

Several systems have been proposed for autonomous human-following tasks. Researchers have contributed to the development of a broad range of studies by addressing various aspects of human-following problems [15].

The face information of people with other features was utilized in visual-based methods. In [16], human recognition was performed by fusing face recognition with skeletal estimates generated by human-following robots. In [17], a mobile robot was equipped with a radio-frequency identification (RFID) reader that could track a target person or another through a passive RFID tag attached to the person or robot. However, the face and tag recognition methods have a limitation in that the distance between the robot and the person must be small for identifying the user’s face or reading the tag. Moreover, a user’s face or tag orientation is not always available in the person-following scenarios.

Linxi and Yunfei [18] utilized AdaBoost to train a binary classifier in outdoor environments based on a sparse point cloud obtained from LiDAR. Cha and Chung [19] applied a one-class classification algorithm called a support vector data description (SVDD) to classify people based on the leg shape by generating feature vectors of the leg segments in a three-dimensional (3D) space from the LiDAR sensor. However, these algorithms do not address cases in which a target is partially occluded by another person.

Chi et al. [14] proposed a gait recognition method using a dataset that records the skeletal joints of people in 3D coordinates using an RGB-D camera to conduct human-following tasks. Stein et al. [20] implemented 24 features to take advantage of the motion of people and improve navigation capabilities.

Many re-identification methods based on target appearance, such as scale-invariant feature transformation (SIFT) [21], color [13,22], and template matching [23] have been proposed. Gupta et al. [23] developed a novel driving algorithm based on a template-matching clothes method using a k-dimensional tree-based classifier with a SURF-based tracker to detect the target appearance and a Kalman filter motion predictor to follow the target person. However, the drawback of this algorithm is that the frame rate is low, approximately six frames per second (fps), and consequently, it has an adverse effect on the computational cost. The method proposed in [21] relies on keypoint-based feature matching to perform data association. However, SIFT-based methods are often not robust against common sources of variation. Furthermore, they have a low frame rate, which drops suddenly with an increase in the number of people in the scene, and keypoints have to face the camera continuously.

A re-identification method for a robot was presented in [24] using thermal image entropy-based sampling to obtain a thermal dictionary for training a support vector machine (SVM) classifier after the head region was segmented for each person. In [25], a person tracking and identification method for a mobile robot was presented by combining three features from two laser range finders (LRFs) and a camera. A person was recognized using illumination-independent features (i.e., gait and height) and color features.

Recently, to achieve accurate and robust object detection and tracking, researchers have employed deep learning [9,13,26,27,28]. Chen et al. [28] used a stereo camera with an Ada-boosting algorithm based on convolutional neural networks (CNNs) for the person tracking using a mobile robot. However, the limitation of this algorithm is that once the selected person walks out of the robot’s field of view (FOV) for an extended period, the neural networks are updated with background data in the scenery because the online CNN model only acts as a feature extractor. Lee et al. [26] applied you only look once (YOLO) as a deep learning technique to detect and track people and a matching method to identify the target person. However, the computational cost of this method was high, approximately around 0.3 s, despite using a graphics processing unit (GPU). Tracking information was also used in [9,13], where identification was based on a Hue, Saturation, and Value (HSV) space to extract color features from clothes and estimate the target person’s position in real time over all frames. In these systems, person identification worked well under moderate illumination changes; however, this approach failed under severe illumination changes. This limitation was resolved by continuously updating the model to accommodate changes over time [22]. However, the updated system depends entirely on color features, which is the main limitation; it leads to failures when more people wear similar clothing. Pang et al. [27] applied an integration method of supervised learning and deep reinforcement learning with a deep Q-network to train an agent and develop a robot that could follow a target person. However, these appearance features become meaningless when other people wear similar clothes to the target person. Thus, this approach is difficult to apply practically, even with a deep neural network, especially in facilities where people wear the same clothes.

## 3. Human Identifier Methodology

Person identification is challenging for mobile robots because of inaccurate bounding box generation, background clutter, occlusions, illumination changes, and unconstrained walking. This results in variations and uncertainties [29]. Figure 1 shows the flowchart of the proposed methodology. Person identification consists of two steps: (1) features extraction and (2) an online boosting algorithm. Online boosting operates by arranging weak learners in a sequence (blue arrows in Figure 1) to build a strong classifier. As the starting point to extract the features used, human detection is required. It is necessary to manually select the desired target person from a live video using a mouse. CNNs have achieved state-of-the-art performance on various visual recognition tasks [30], such as image classification [31], object detection [32], and semantic segmentation [33]. However, some of the deep learning techniques have improved dramatically the performance of object detection in real-time videos, such as single-shot detector (SSD) [34], YOLO [35], and mask R-CNN [36]. The proposed system uses an SSD with MobileNets [37]. The details of SSD and MobileNet are beyond the scope of this study. In this section, we briefly explain the features and online boosting-based person classifier used in this context.

### 3.1. Feature Definitions

The primary purpose of the identification model is to establish whether the observed person is the target over successive frames; this involves labeling the observed people as either target or non-target persons. Labeling people without prior knowledge is a fundamental problem in human-following robot systems. Person identification poses an additional challenge when many people wear similar clothing. For instance, it is difficult to distinguish people wearing similar t-shirts without considering other features, in addition to their appearance features. Our model addresses this challenge through online learning that merges many features into a single feature with continuous updating of the features used. The four major features (color, height, location and IoU) are updated online and used as inputs for the online boosting algorithm after applying the normalization technique. The min–max normalization was applied to all features to ensure that the result falls within the range of 0 and 1 [38]. The features used were extracted in this work based on a feature perspective [15]. These features are described in detail in the following sections:

#### 3.1.1. Color Feature

The target person is recognized using an appearance model. Appearances are commonly used to identify the target person in person-following robot systems. We used an HSV color space, which is one of the most popular methods owing to its simplicity and robustness for extracting color features, with the proposed method and an online color-based identification update [22]. The boxes in Figure 2 represent human detection of the deep learning-based model. The yellow and green boundary boxes represent the target person in the previous and current frames, respectively, whereas the red boundary boxes represent other people in the current frame. To extract the color features only from the clothes of the upper body and ignore the rest of the scene, a region of interest (ROI) was applied. Segmentation is a powerful technique in computer vision technology for lowering computing costs [39]. The importance of segmentation increases when a task is performed in real time. The blue boxes indicate the ROIs on the upper bodies of people, while the white contour indicates the color extraction from people’s clothing within ROIs. To normalize the color feature, we used the area ratio given by:(1)$$area\;ratio=\frac{A_{c}}{A_{ROI}}$$
where $A_{c}$ is the area of the contour (white contour) calculated using the method in the OpenCV library created for this purpose, and $A_{ROI}$ is the area of the ROI (blue box). More details on how to extract the color feature can be found in [22], which is beyond the scope of this work.

The vertices of the bounding boxes relative to the entire image resolution are provided in the following formats: ($u_{max},\;u_{min},\;v_{max},\;v_{min}$). The centers of these boxes on the *u* and *v* axes in the image space are as follows:(2)$$\begin{array}{c} c_{u_{i}}=\left(u_{max_{i}}-u_{min_{i}}\right)/2\\ c_{v_{i}}=\left(v_{max_{i}}-v_{min_{i}}\right)/2 \end{array}$$
where i=1,2,…,n denotes the number of people within the FOV of the camera.

#### 3.1.2. Height Feature

A person’s height can be used as another feature for identifying people. Particularly, when many people have similar appearances, height helps to reduce the number of candidates considered for identifying the target person. Some early methods used to estimate people’s heights include those by De et al. [40], who estimated the height of subjects in video surveillance systems based on significant points in a scene as a reference for the system. Hoogeboom et al. [41] estimated a person’s height using a reference height and other criteria, such as the target individual being at the center of the image. However, it is impossible to use a reference height in human-following robot applications; therefore, these methods have limited practical applications. Recently, with the rapid development of applications that utilize depth cameras and computer vision technology, methods have been proposed to estimate the distance and height of people without requiring reference measurements. One of the most popular sensors is an RGB-D camera. To estimate people’s heights, we first need to calculate and determine the following three parameters: (1) measure the distances between people and the camera, (2) calculate the vertical angles at the top of the head region relative to the camera level, and (3) determine the camera height relative to the ground plane. In robotic applications, estimating accurate distances in 2D image space is insufficient. Therefore, distance estimation in 3D space is indispensable. The distances are measured directly using an RGB-D sensor after determining the center points of the objects relative to the camera position using a point cloud [42]. The distances $d_{i}$ from the camera pose to $person_{i}$ (in meters) are defined as:(3)$$d_{i}=\sqrt{x_{i}^{2}+y_{i}^{2}+z_{i}^{2}}.$$

The mobile robot was equipped with an Orbbec Astra camera that provided synchronized color and depth data at a resolution of 640×480 over a $49.5^{\circ }$ vertical FOV and $60^{\circ }$ horizontal FOV, as shown in Figure 2. The height of the camera was 147 cm from the ground plane to obtain a better view of the environment, as depicted in Figure 3. The angles of people’s location in the image space relative to the center of the images are dependent on the sensor specifications used, which are given on the $\theta_{h}$-axis and $-\theta_{v}$-axis as follows:(4)$$\begin{array}{c} \theta_{hi}=-0.09375\times c_{u_{i}}+30^{\circ }\\ \theta_{vi}=-0.1031\times v_{min_{i}}+24.75^{\circ } \end{array}$$
where $\theta_{hi}$ is the horizontal angle of a person at the body center relative to the image center and $\theta_{vi}$ is the vertical angle of a person at the topmost position (i.e., top of the head region) relative to the camera level. $c_{u_{i}}$ was calculated using Equation (Equation 2).

Once these three parameters are known, the heights $\left(h_{i}\right)$ can be obtained (in centimeters) as follows:(5)$$h_{i}=d_{i}\times 100\times \tan\left(\theta_{vi}\right)+147.$$

To improve the robustness of the height feature, we use the difference in height instead of the actual height because the height feature is sensitive to the continuous displacement of the upper body in the up-and-down direction due to the person’s walking and the movement of the robot when following the person. Conversely, the height difference helps the model deal with challenging situations, such as the up-and-down displacement of the upper body when walking, which is given by:(6)$$dh_{i}=\left|h_{tl}-h_{i}\right|$$
where $h_{tl}$ is the estimated height of the target person in the last tracked frame and $h_{i}$ is the height of the people in the current frame.

To normalize the height difference, we assume that the minimum and maximum height differences are 0 and 20 cm in absolute value, respectively, which are given by:(7)$${{dh}_{i}}^{*}=1-\left(\frac{dh_{i}-\min}{\max-\min}\right)=1-\left(\frac{dh_{i}}{20}\right).$$

If a person with a height difference greater than 20 cm is present around the target person, the model considers their height difference as the maximum difference.

#### 3.1.3. Localization Feature

Another feature of person identification is the localization of people in an image. This feature is also useful in reducing the number of candidates considered for a target person when many people have the same height and similar appearance. In this work, localization refers to a person’s position in the image space on the $\theta_{h}$-axis. To calculate the position of a person in the image space, we used the horizontal angle in Equation (Equation 4) Although the robot attempts to maintain the orientation of the target person in the heading direction, that is, the position of the target person at the center of the image, when the target person turns left or right, this center-image position is not maintained. However, we use the difference in angle instead of the angle itself, which is given by:(8)$$d\theta_{hi}=\left|\theta_{htl}-\theta_{hi}\right|$$
where $\theta_{htl}$ is the measured horizontal angle of the target person in the last tracked frame and $\theta_{hi}$ is the horizontal angle of people in the current frame, which are obtained from Equation (Equation 4). The horizontal FOV of the sensor used between the image center and the far right or left is $30^{\circ }$, as described in Equation (Equation 4) and Figure 2. To normalize the horizontal angle difference, we assume that the minimum and maximum of the angle difference are $0^{\circ }$ and $30^{\circ }$ in absolute value, respectively, which are given by:(9)$$d\theta_{hi}^{*}=1-\left(\frac{d\theta_{hi}-\min}{\max-\min}\right)=1-\left(\frac{d\theta_{hi}}{30^{\circ }}\right).$$

Assume that there is a person with a horizontal angle difference greater than $30^{\circ }$ around the target person: for instance, the target person on the left side or other people on the right side. In this case, the system considers the horizontal angle difference as the maximum angle.

#### 3.1.4. IoU Feature

IoU represents the area ratio of the intersection to the union of two shapes, for example, boundary boxes [43]. We observed that IoU sometimes drops suddenly to less than 0.5 because the size of the boundary box is minimized or maximized in some situations, that is, when another person partially occludes the target person or for some other event. However, we modified the denominator to avoid this situation, which presents the same IoU result before modification when both boundary boxes are almost identical; we define the modified IoU in this work as follows:(10)$$mIoUi=\frac{|Ai\cap B_{tl}|}{min(Ai,B_{tl})}$$
where $B_{tl}$ is the last boundary box of the target person and Ai is the current boundary box of people, including the target person. Figure 2 shows the boundary boxes of people in the last and current frames.

Using height, localization, and IoU, the features of the people in the current frame are compared with those of the target person in the previous frame to improve the identification performance. The values of all features are between 0 and 1; these values are applied to the online boosting model, as explained below.

### 3.2. Online Boosting-Based Person Classifier

Boosting is a popular and powerful ensemble learning technique [44]. Traditional weakly supervised learning algorithms classify examples [45,46] based on a single model, such as naive Bayes or neural networks. Ensemble classifiers build a strong classifier by combining many weak classifier-based models, each of which is learned using a traditional algorithm to improve the performance of the learning method [47]. Contrarily, boosting is a more complex process that generates a series of base models $h_{1},\;h_{2},\;\ldots ,\;h_{N}$. Each base model $h_{N}$ is learned from a weighted training set whose weights are determined by the classification errors of the preceding model $h_{N-1}$ [48]. Many ensemble learning studies that use offline [49] and online [50] boosting algorithms have been proposed over the years. Online boosting algorithms are primarily used in self-learning applications [51]. Such algorithms have advantages over typical offline algorithms in applications where data continuously arrive. As an ensemble model, the boosting model comes with an easy-to-read and interpret algorithm, making its prediction interpretations easy to handle. Boosting is a resilient method that curbs over-fitting easily [52]. The boosting model quickly also adapts to abnormal conditions and improves the performance of the applications, which receive data in real time [53].

## 4. Results and Discussion

### 4.1. Online Boosting Algorithms Evaluation

#### 4.1.1. Dataset Preprocessing

There are two important factors to be considered while setting up online boosting. First, weak learners must be online algorithms. Second, the number of ensemble weak learners must be specified prior to training. A weak classifier is an incremental learning algorithm that takes the current hypothesis and training example as input and returns an updated hypothesis [48]. We compared a wide variety of weak classifiers in terms of accuracy and speed. To achieve this comparison, labeled data must be used. We evaluated the performance of four weak learners: perceptron (P), decision stump (DS), decision tree (DT), and naive Bayes (NB) classifiers using the iris dataset (https://archive.ics.uci.edu/ml/datasets/iris accessed on 17 September 2022 ) after processing the streaming samples individually, that is, the samples fed the models one by one. The dataset contains 150 samples divided into training and test data. The test data size was set to 30% (45 samples), while the remaining 70% (105 samples) were randomly selected from the original dataset for training. In offline learning, the training and test data were input into the model all at once. In contrast, in online learning, the data were fed into the model one by one. The first sample from the training data was input into the model, and the model was tested for all test data samples. Subsequently, the process was continued until the final sample was obtained. The total number of model tests was 4725 times (105 samples for training × 45 samples for testing).

#### 4.1.2. Performance Metrics

Figure 4 and Figure 5 show comparisons of the accuracy and computation time for all weak learners, respectively. The x-axis indicates the number of training samples, and the y-axis indicates the cumulative average accuracy in Figure 4 and the cumulative average computation time in Figure 5. As observed, the accuracy of the decision stump was approximately one after ten training samples, while the accuracy of the remaining models became approximately one after training 30 samples.

The cumulative average computation times of perceptron, decision tree, naive Bayes, and decision stump algorithms were 0.234, 0.224, 0.384, and 0.006 ms, respectively. Remarkably, the decision stump was approximately 39, 37, and 64 times faster than the perceptron, decision tree, and naive Bayes algorithms, respectively. The computation time of the decision tree increased with an increase in the number of training samples. To minimize computation time and achieve good accuracy, we ultimately selected the decision stump model as a weak learner for our online boosting algorithms.

Many online boosting algorithms have been developed, such as online adaptive boosting called OzaBoost (OZaB) [48], online gradientboost (OGB) [51], online smooth-boost (OSB), online smooth-boost using online convex programming (OSB.OCP), and online smooth-boost with prediction with expert advice (OSB.EXP) [44]. Before comparing the accuracy of the online boosting algorithms, as in the case of weak learners, we must first select the appropriate number of weak learners to be used. In this study, we compared the performance of different online algorithms by increasing the number of weak learners (decision stumps) as follows [1, 5, 10, 20, …, 140, 150], as shown in Figure 6 and Figure 7.

Figure 6 shows the relationship between the number of weak learners and computation time for all online boosting algorithms. The number of weak learners is directly proportional to the computation time. The OzaBoost algorithm was the fastest in all cases, whereas gradientboost was the slowest algorithm, especially when the number of weak learners was greater than 70. In the gradientboost algorithm, the number of selectors (K) must be chosen beforehand, which is primarily used for feature selection [54]. This study considers K=1 for a fair comparison.

As shown in Figure 7, all algorithms achieved accuracies between 0.975% and 1.0%, except for the OSB.OCP algorithm. Therefore, we set up 40 weak learners with which we obtained the best performance for all algorithms to evaluate the accuracy of the five online boosting algorithms with increasing training samples.

Figure 8 shows an accuracy comparison with an increasing number of training samples. All the algorithms achieved high accuracy after training for almost 30 samples. The x-axis represents the number of training samples, while the y-axis represents the cumulative average accuracy of the online boosting algorithms. The performance of all boosting algorithms consistently improved with the continued feeding of the model by the training samples.

The aforementioned discussion is a simplified analysis of various online boosting algorithms. There are other weak learners, datasets, and boosting algorithms that are not included here owing to space constraints. Among the algorithms with high accuracy, we selected the OzaBoost algorithm, which was the fastest for our proposed system. Then, the quality of the model is further defined by performance metrics, including precision, recall, and F1 score [55]. All of them were equal to 0.978. Precision is given by:(11)$$\;\mathrm{Precision}\;=\frac{TP}{TP+FP}.$$

Recall is given by:(12)$$\;\mathrm{Recall}\;=\frac{TP}{TP+FN}.$$

*F*-measure is given by:(13)$$F-\;\mathrm{measure}\;=\frac{2\times \;\mathrm{precision}\;\times \;\mathrm{recall}\;}{\;\mathrm{precision}\;+\;\mathrm{recall}\;}.$$
where TP,FN and FP are the number of true positives, false negatives and false positives, respectively.

Applying an online boosting algorithm to the proposed system requires initial training to label people as target or non-target people. We employ the aforementioned four features to recognize the target person: color ($f_{1}$), height difference ($f_{2}$), localization difference ($f_{3}$), and IoU ($f_{4}$). Features $f=[f_{1},f_{2},f_{3},f_{4}]$ have values between 0 and 1. In the ideal case, the values of the features are f=[1,1,1,1] for the target person (y=1) and f=[0,0,0,0] for the non-target person (y=−1); however, practically, it was not easy to differentiate them. We assumed that if the color feature value is greater than 0.3 and if the height and localization differences are greater than 0.7 and 0.85, respectively, and if the IoU is greater than 0.5, they should be the target person, that is, f≥[0.3,0.7,0.85,0.5]. Otherwise, they should be a non-target person. Those values were empirically set as the thresholds in our study. However, if the illumination is more uniform in the working environment, it is better to increase the color threshold to 0.5 or more, and if the height differences between the target and other people are large, it is better to increase the threshold of the height differences to 0.8 or more. In addition, if the robot follows the target in a straight line, it is better to increase the threshold of the localization differences to 0.95 or more, and if there are no occlusion situations for the target, it is better to increase the IoU threshold to 0.75 or more. Otherwise, it is better to decrease them based on the environment and path conditions. These assumptions label people around the robot and generalize the proposed model. Therefore, our system does not require any prior information regarding the target person, regardless of the color of their clothes or their height. The target person to be followed was manually selected using the mouse, as mentioned earlier.

In the initialization, the system randomly generates 100 labeled samples for the target person and 100 others for non-target people according to our assumptions before selecting the target person to guarantee that the model is ready. The system continued learning based on people’s information after selecting the target, as long as the system was running.

### 4.2. Infrastructure Setting

#### 4.2.1. Platform

In this study, we used a differential mobile robot called Rabbot manufactured by Gaitech, as shown in Figure 3. Rabbot weighed 20 kg and was designed to carry a load of up to 50 kg. Consequently, a high frame rate was required for smooth movement. The robot was equipped with a SLAMTEC RPLiDAR A2M8 to protect itself from collisions, an Orbbec Astra Camera for tracking people, and an onboard computer (hex-core, 2.8 GHz, 4 GHz turbo frequency i5 processor, 8 GB RAM, and 120 GB SSD) in addition to a computer at a workstation (Intel Core i7-6700 CPU (Central Processing Unit) @ 3.40 GHz). Both computers ran under robot operating system (ROS) Kinetic+Ubuntu 16.04 64-bit.

#### 4.2.2. Environment

A realistic scenario of the testing environment is illustrated in Figure 9. The black dashed line indicates the path of the robot and target person in the testing environment. The path starts from the Helper’s laboratory to the end of the corridor. The black, green, and red circles represent the robot, the target person, and other people, respectively. Other people wore t-shirts that are the same color as the target’s t-shirt. Three people wore the same t-shirt including the target person. The heights of the people were 175, 185, and 173 cm, correspondingly referred to persons A, B, and C.

However, the operating environment was narrow. Many researchers in other laboratories walked into the area during the experiments wearing normal clothes. The blue letters denote the glass walls and windows at the corridor ends.

### 4.3. Human-Following Experiments

We conducted extensive experiments using three different colors (black, white, and blue) to evaluate and compare the performance of the proposed person-identification framework. This framework was proposed to identify the target person based on four features. These features were combined into a joint feature and learned using the online OzaBoost model. We divided our experiments into two categories in the case of black. The first category of experiments adopted all features as the remaining colors. Contrastingly, the second category adopted only two features, color and height, to compare our system with the previous significant system, as shown in Table 1.

In our experiments, the target person and others had the same appearance. A video of the mobile robot following the target can be found at the following link: (https://www.youtube.com/watch?v=jJaM1D6-EdM accessed on 17 September 2022).

In the following subsections, we describe these experiments in more detail.

#### 4.3.1. Human-Following Experiments Using Four Features

In this section, the experiments conducted to evaluate the system performance based on four features using three different colors are described. The experimental results for the blue, white, and black are summarized in Table 2, Table 3 and Table 4, respectively. These tables represent the summary of experimental results as part of our experiments to demonstrate the experiment’s status, travel distance, travel time, number of frames, average speed of the robot, successful and failed tracking rate of the entire system, online boosting model, fps, and so on. The proposed system also recorded the number of frames lost. There are two types of lost frames: The first type is a lost frame of the target person in the model, that is, the online boosting model owing to incorrect height estimation or other reasons. In this type, the online boosting model is fed data, and the model considers the target person to be a non-target. We calculated the successful tracking rate for the model based on the number of frames provided to the online boosting model. Mathematically, the successful tracking rate is computed as follows:(14)$$\;\mathrm{Successful}\;\mathrm{Tracking}\;\mathrm{Rate}\;=\frac{n}{N}\times 100$$
where *N* is the total number of frames (No. of frames) and n is the number of successfully tracked frames for the target by the tracking algorithm [23]. In the second type, the camera detects people in RGB data, but no depth data are available to estimate the height owing to the noise in the sensor itself. For this type, there are no inputs or outputs for some frames in the online boosting model, i.e., there are some frames lost due to noise, which are counted by ($N-N_{m}$), where $N_{m}$ is the number of frames for model. Mathematically, the successful tracking rate for the model is computed as follows:(15)$$\;\mathrm{Successful}\;\mathrm{Tracking}\;\mathrm{Rate}\;\mathrm{for}\;\mathrm{Model}=\frac{n}{N_{m}}\times 100.$$

Based on all the frames of the system, we calculated the successful tracking rate for the entire system. Consequently, the successful tracking rate for the model was greater than or equal to the successful tracking rate for the entire system. The symbols (O, X) in the second row of all the tables refer to the experimental status in which O refers to a successful experiment, whereas X refers to a failed one. In our subjective assessment, we judged that the experiment was successful if the mobile robot arrived at the destination point for the target person known beforehand and failed otherwise, regardless of the travel distance in the failed experiments.

Persons A, B, and C were the leaders wearing blue, white, and black t-shirts, respectively, during testing. Thirteen experiments were performed for each color, as shown in the three tables. Overall, the mobile robot arrived at its destination in all the experiments involving the blue t-shirt. In contrast, it arrived at 12 and 11 experiments involving white and black, respectively. We consider this failure to be due to the limitation of online boosting, which is sensitive to noise and outliers, thus creating a bias in the predictions, as reported in [56]. In all experiments, the average number of frames per second of the proposed system was greater than 24, i.e., 41.66 ms, which is suitable for making the robot movement smooth and compatible with the frame rate of the camera using only the CPU.

We selected one experiment from each group to show color extraction, IoU, height estimation, height difference, localization, localization difference, and their normalization as plots. As mentioned earlier, three participants wore the same t-shirts in all the experiments. One was a target person and two were non-targets.

In the blue t-shirt case, person A was the target with a height of 175 cm, while persons B and C were the targets for the white and black t-shirt experiments with heights of 185 and 173 cm, respectively. Figure 10a–c show the height estimation, height difference, and normalization of the height difference of the target person over all frames, respectively, while the robot follows the target person. The blue, green, and black curves represent persons A, B, and C, respectively. In the beginning, the height estimation is almost constant when the person does not walk in the first frames and then varies up and down owing to the person walking and the robot’s movement. To resolve this issue, we used height difference. The height difference method also helps the system deal with the up-and-down displacement of the upper body while walking, which is impossible to solve using the absolute height. The height difference between the height of the target person in the current frame and that in the last tracked frame for the majority of the frames was less than 4 cm. However, some values were greater than 4 cm and less than 7 cm, as shown in Figure 10b. Therefore, the height feature was not robust when the height difference between the target and non-target heights was less than 7 cm. This height difference range may decrease or increase with other sensors. We only considered the height difference as an aid in reducing the number of candidates for the target person.

Figure 11 shows the IoU of the target person across all the frames. For most frames, the IoUs of persons A, B, and C were greater than 0.90, 0.93, and 0.85, respectively. People’s walking and clothing color with the background scenery play a role in determining the value of IoU. However, the IoU is the most robust feature in this study because of its high value for the target person and its low value for other people; that is, its value is approximately zero unless partial or complete occlusion occurs.

Figure 12a–c depict the localization, localization difference, and normalization of the localization difference of the target person over all the frames, respectively. As aforementioned, the horizontal FOV of the camera was $60^{{}^{\circ}}$.

Generally, the target person has located in the $+\theta_{h}$ direction of the images when he tries to walk on the robot’s left side, while the target is located in the $-\theta_{h}$ direction when he tries to walk on the right side, as shown in Figure 2. Initially, all the target persons were located at the center of the images before walking. Persons A and C were located in the $+\theta_{h}$ direction of the images, while person B was located in the $-\theta_{h}$ direction after walking for some frames. Although the robot tries to maintain its target in the heading direction to an extent, the horizontal angle is not approximately zero in all frames, particularly when the target turns left or right. This implies that the horizontal angle should be approximately zero. The minimum angles were approximate $-18.2^{\circ }$, $-15.2^{\circ }$, and $-19.9^{\circ }$ for persons A, B, and C, respectively, when the people moved out of the laboratory and turned right at the door to continue walking in the corridor. The maximum angle was less than $10^{\circ }$ for all people, as shown in Figure 12a. However, the localization difference was less than $3^{\circ }$ for all people, which is relatively small compared with the entire horizontal FOV of the camera used, as depicted in Figure 12b. As observed, the robot moved smoothly when person B was a target compared to other people.

Figure 13 illustrates the area ratio of the color feature across all the frames. People A, B, and C wore blue, white, and black t-shirts, respectively, as targets during the experiments. The area ratio for the color feature was close to 1 when the target was walking in the laboratory for all colors because of relatively uniform illumination, while it dropped to 0.3 in some frames where person B was walking in the corridor (i.e., from 330 frames to the end of the experiment) in case of white owing to non-uniform illumination. Black maintained a high area ratio, whereas white did not because the color of the clothes tended to be black in the corridor owing to illumination changes. The color feature is meaningless when other people wear similar clothes, which is the main goal of this study. However, it was helpful in reducing the number of candidates for the target person when other people were non-volunteers in these experiments, wearing different clothes and moving around the target person.

The goal of normalization is to change the values of the height difference, localization differences, and color features to a common scale without distorting the differences in the ranges of values. Figure 14 shows the normalization features of people. The blue, green, and black curves represent person A as the target person and persons B and C as non-target persons, respectively, when they all wore blue t-shirts. The robot started detecting another person as the first non-target person at frame number 466 until frame number 610, while the target person completely occluded the second person during walking. The second person was detected as a non-target person at frame number 666 until 759 after the target person turned right slightly to move next to the second one. Simultaneously, the first one was behind the robot and the target person owing to the movement, as shown in Figure 9.

The normalization process allows us to evaluate the importance and stability of every feature, that is, whether its range is narrow or wide. Although the target did not walk straight during the experiment, the localization feature had the smallest range, particularly after 400 frames, when walking in the corridor (Figure 14a). The localization of other people (persons B and C) in the image space was near the target localization (person A) around the center of the images when they were far from the robot and then moved to the top right and left when the robot was very close. The IoU feature is the most robust because of the significant difference between the IoU of the target and that of people, which were approximately 1 and 0, respectively, for the majority of frames (Figure 14b). Some intersections occurred; thus, the IoU of other people was between 0 and 1 in some frames (Figure 15i–l). The height feature had the widest range compared to the others owing to the robot movement and the up-and-down displacement of the upper body while walking (Figure 14c). The height feature is acceptable for person B because of the large height difference between the target person and person B, whereas it is weak for person C in some frames because of the small height difference between the target person and person C. The differences were 10 cm and 2 cm for persons B and C, respectively. The color feature is meaningless in this work based on our assumption that X≥[0.3,0.7,0.85,0.5]. The value of the feature was greater than 0.3 for all people (Figure 14d). However, we conducted several experiments using only two features in Section 4.3.2 compared with four features to evaluate the performance of the proposed framework based on the features used.

Figure 15 displays snapshots captured by the system. At the beginning of the test, person A, who wore a blue t-shirt, stood in front of the mobile robot, where the red box indicates that the person was non-target (Figure 15a). The user who operated the system selected him as the target using the mouse (see the yellow box in Figure 15b), where the yellow box represents the last track frame until the end of the experiment. After selecting the target person, the red box was changed to the green box, where the green box indicates that person A has become the target that the robot should follow (Figure 15c). At this moment, the target person carried the joystick to stop the robot by pressing a button on it until the mobile phone camera was ready to record a video showing the robot’s behavior, which was on the right side (Figure 15c). When the mobile phone camera was ready to record the video, the person started walking, and the robot began following him. The mobile robot followed the target person from the laboratory as the departure point to the end of the corridor as the destination point, as shown in Figure 9. When walking through the corridor, there were two volunteers; one was standing on the right side in the middle of the corridor (Figure 9), where the robot was detected as a non-target (Figure 15d). After a few meters, another person standing on the left side (Figure 9) was also detected as a non-target (Figure 15e). Both volunteers wore t-shirts similar to the target person, who wore blue t-shirts in this experiment. During the experiment, the target person was occluded by another person when she attempted to move between the target person and the robot to go to her laboratory from the left side (Figure 15f) to the right side (Figure 15g), and the robot lost the target tracking in two or three frames when the occlusion was complete. However, the robot tracked the target person when the occlusion was partial (Figure 15f) and then correctly re-identified him with the online person identification model once he partially reappeared in the camera view (Figure 15g). Target re-identification was fast and robust owing to the combination of multiple features using the online boosting model. The modified IoU remarkably improved the identification model to quickly identify and re-identify the target when the box was minimized or maximized suddenly, such as in partial occlusion situations. The robot continued to succeed until it arrived at its destination (Figure 15h).

In the white and black t-shirt experiments, the robot followed other targets in a similar manner as in the blue t-shirt experiments, and the volunteers stood at approximately the same spots for a fair comparison. Figure 15i,j show person B wearing the white t-shirt as a target and two persons standing up in the middle of the corridor as non-targets. One was on the right side of the target person and the other was on the left side, similar to Figure 15d,e in the blue t-shirt experiment. Person C walked wearing a black t-shirt next to people who stood up to his right Figure 15k and left Figure 15l in the middle of the corridor, which was similar to Figure 15d,e in the blue t-shirt experiment.

The last three snapshots in Figure 15 show different experiments beyond the scope of the three aforementioned Table 2, Table 3 and Table 4. Figure 15m shows that person B and person C walked side-by-side in the same direction when the robot followed person B as a target, whereas Figure 15n shows that they walked side-by-side in opposite directions when both of them wore white t-shirts. We conducted many experiments in which the two persons standing next to each other and the target person passed through the center (Figure 15o). All people wore black t-shirts, including the target person. In this experiment, person C was the target of a mobile robot. Although significant illumination changes occurred during the experiments, the color feature was continuously updated to accommodate these changes over time. All experiments above were conducted in the late evening.

Figure 16 shows the capability of target identification under a different lighting environment. We perform extra experiments in the early morning when the sunlight passes into the corridor while it does not pass into the lab at all. On the other hand, the illumination is approximately uniform in the lab (Figure 16a), while it is non-uniform in the corridor (Figure 16b–d). As one can see from the snapshots in the corridor, the white T-shirt tends to be darker white because of non-uniform illumination in the corridor due to sunlight, which passed from the glass windows. Overall, the mobile robot successfully followed its target in most experiments, as described herein.

#### 4.3.2. Comparison with Previous System: Using Two Features

This section describes the experiments performed to compare the proposed system with the previous method, which is closely related to our study, based on the features used. Unlike previous methods in related works, Koide et al. [3] introduced a tracking system using OpenPose with human height estimation relative to the ground plane prior information. The appearance features of people were extracted based on a combination of convolutional channel features and merged with height using online boosting to identify the target person. This method leverages a deep learning model to extract the appearance features and an online boosting ensemble model, which ensembles many weak classifiers to build a strong classifier. This online boosting model requires selectors for feature selection. Naive Bayes was adopted as a weak classifier, and the total number of weak learners was 150; however, the researchers did not explain how this number was selected as we did. Moreover, this method mainly depends on two features: height and appearance. It also leads to failures when more people have similar appearances and the same or similar heights. The mobile robot (Pioneer P3AT) followed a target in both outdoor and indoor environments and was equipped with an NVIDIA Jetson TX2 and a monocular camera in their work. Comparison with other related work in the human-following system for the overall system is relatively difficult owing to several reasons, such as different platforms, different sensors/hardware, non-identical operating conditions, and reported results. Nevertheless, a comparison at the individual module level is possible [23].

We focused on the personal identification model in this study, which is considered an essential model for a robot to follow a specific person when there are other people around it. Therefore, we conducted 13 experiments based on the two features used in the previous approach to identify the target person for comparing the results with the four features used in our study as an indirect comparison. The results of these experiments are summarized in Table 5. Person C and others wore the same black t-shirts, similar to the experiments involving four features, as shown in Table 4. The symbol X of the second row in the table refers to failed experiments. The robot failed to arrive at the destination point in five experiments with two features, as listed in Table 5. In comparison, it failed in only two experiments for the four features, as shown in Table 4.

Figure 17 displays snapshots captured by the system during experiences with the two features. The robot correctly tracked the target person from the laboratory until he arrived next to another person (Figure 17a) and tracked another person as a target person in the few frames (Figure 17b). It then corrected its decision to track the target (Figure 17c) and tracked another person again as the target (Figure 17d). Finally, the robot failed completely (Figure 17e). In this experiment, the robot was confused and captured frame shots in all directions. However, the robot arrived at the destination point in eight experiments, although it tracked another person as a target in a few frames before returning to correct its mistake and then continued following it, as shown in Figure 17. We did not observe the robot tracking another person as a target and then returning to correct its decision in the case of the four features, owing to the nature of the features used. Thus, we consider not correcting the decision as a limitation of the proposed person identification model and expect that using the average of the last three or four frames instead of only the last one may solve this limitation as well as improve the outcome. This failure and tracking of the wrong person were expected because of the difficulty of the task, which requires more features to help the robot follow its target efficiently.

In summary, regardless of the type of algorithm used for human detection, color extraction, and height estimation, many people wear the same t-shirts and have the same or similar height. In this case, the system fails to identify the target efficiently unless an extra feature helps distinguish between the target and non-targets. Nonetheless, we can say that the localization and IoU features play a significant role in improving the proposed system. Generally, the tracking performance of the proposed person identification model is better than or similar to that of state-of-the-art models. The experimental results show that using the proposed approach leads to promising results.

## 5. Conclusions

This study presents a multi-feature framework in which we integrated four features using an online boosting approach for a human-following robot. The proposed framework leverages a deep learning technique to detect and track people in a robot space. We presented a novel person-identifying model to identify a target person in a challenging situation in which people around the target person wear identical or similar clothes and have the same or similar height. The person identification model extracts color features, estimates the height and location, and calculates the IoU. These features were combined into a joint feature using the online OzaBoost algorithm after comprehensively evaluating several online boosting algorithms with the OzaBoost algorithm in terms of accuracy and speed. Furthermore, it continuously updates these features in all frames to identify the target person efficiently. The experiment proved that the proposed model is a generalized model that can be applied to anyone without prior knowledge, regardless of their appearance and height. Through evaluations based on the features used, it was demonstrated that the proposed identification model outperformed other state-of-the-art models.

Although the proposed model demonstrated some limitations, such as not being able to correct the decision when tracking the wrong person, it has promising applications in mobile robots, which follow dynamic objects to provide personal assistance and service and assist in large storage and manufacturing industries.

In future work, we plan to incorporate a camera that can capture less noisy data and adds more features to improve the tracking success rate, making the process more efficient. Moreover, we also plan to improve the proposed system to follow the target person using a multi-robot.

## Figures and Tables

**Figure 1 sensors-22-08422-f001:**
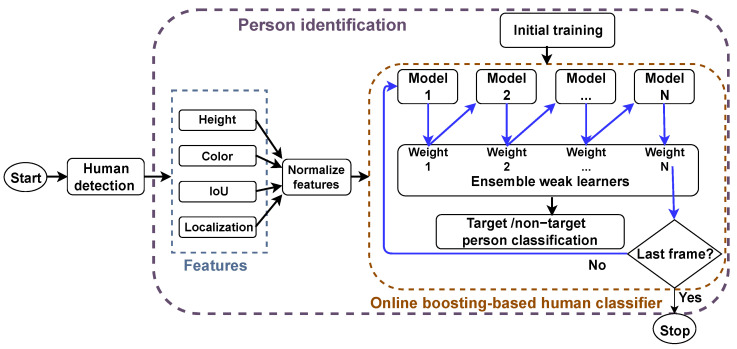
Flowchart of the proposed system.

**Figure 2 sensors-22-08422-f002:**
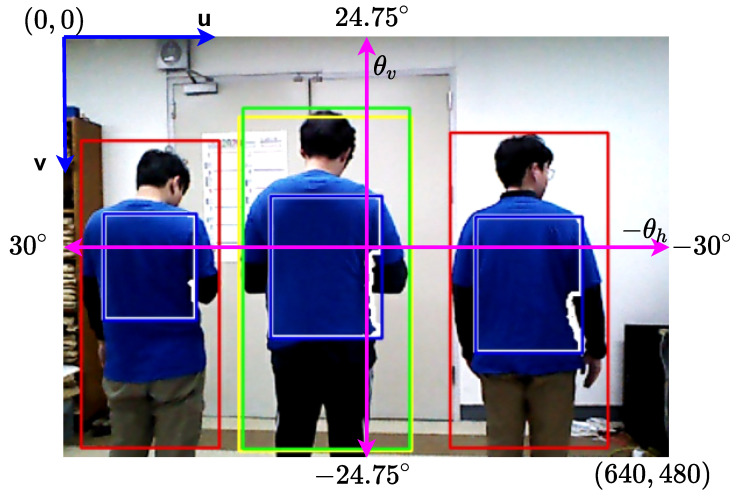
Human detection and color extraction within the region of interest.

**Figure 3 sensors-22-08422-f003:**
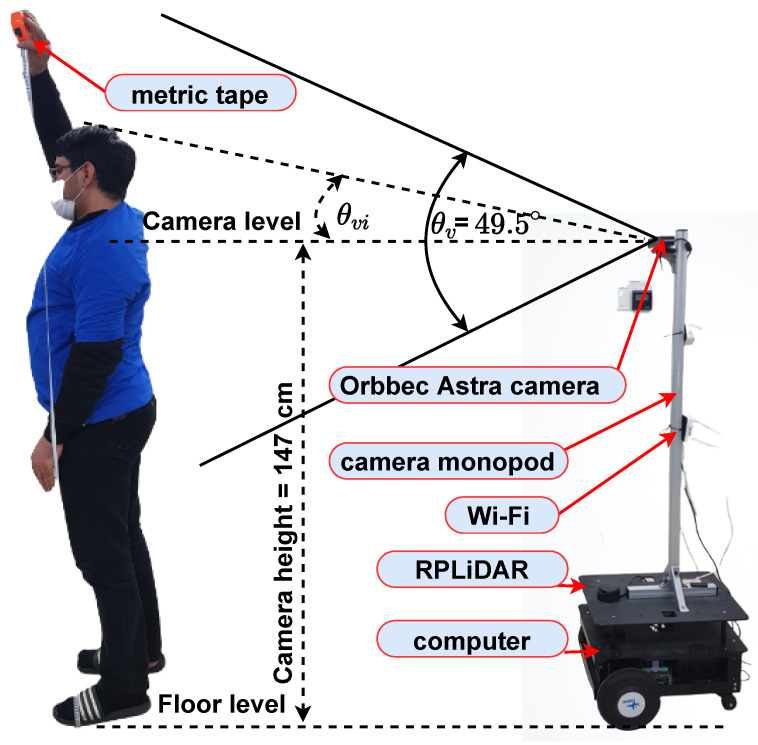
Mobile robot mounted with an RGB-D camera and necessary sensors.

**Figure 4 sensors-22-08422-f004:**
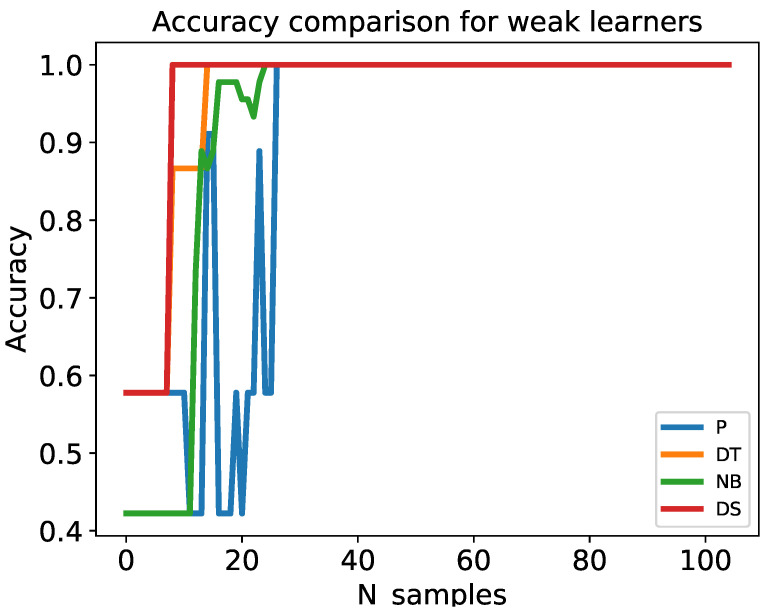
Comparison of weak learner accuracy.

**Figure 5 sensors-22-08422-f005:**
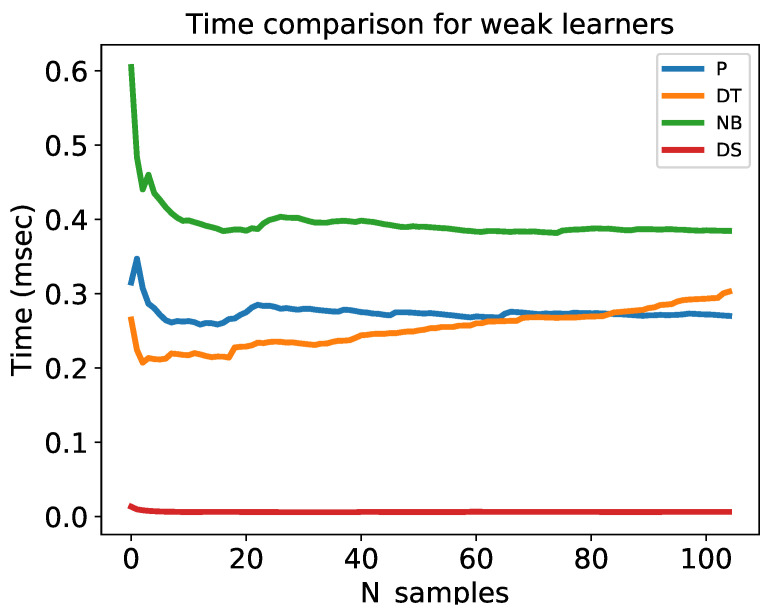
Comparison of computational time for the weak learner.

**Figure 6 sensors-22-08422-f006:**
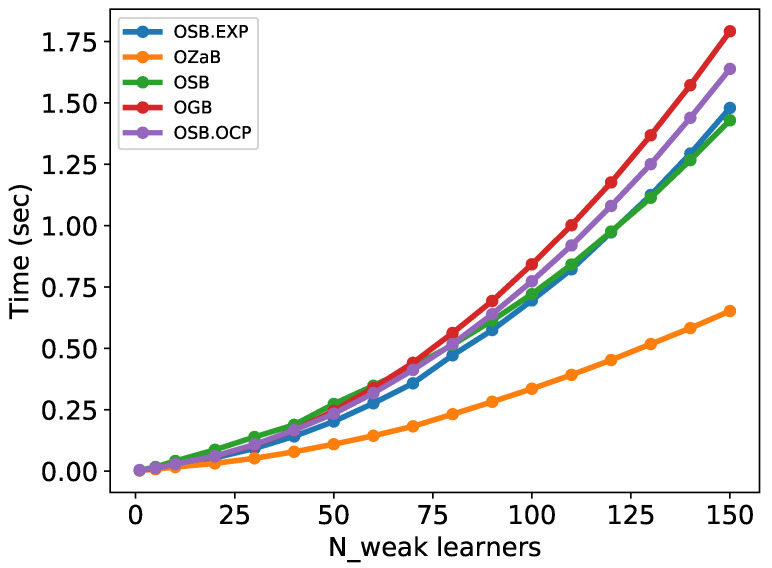
Comparison of computational time for online boosting algorithms.

**Figure 7 sensors-22-08422-f007:**
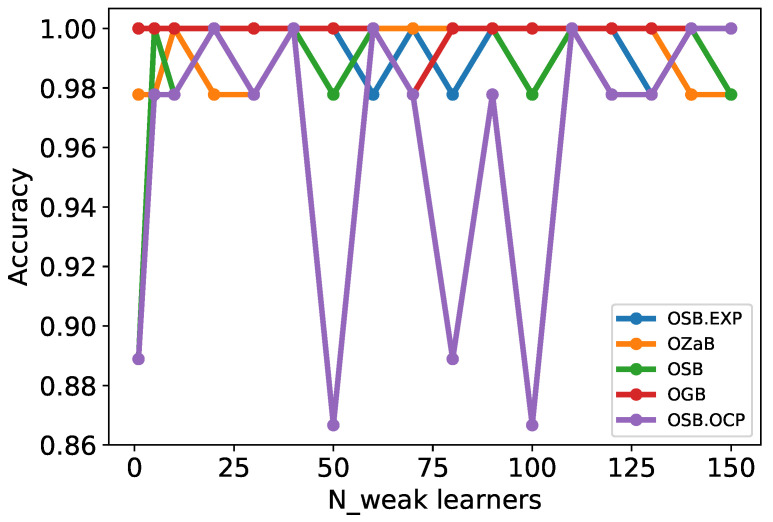
Comparison of online boosting accuracy with the number of weak learners.

**Figure 8 sensors-22-08422-f008:**
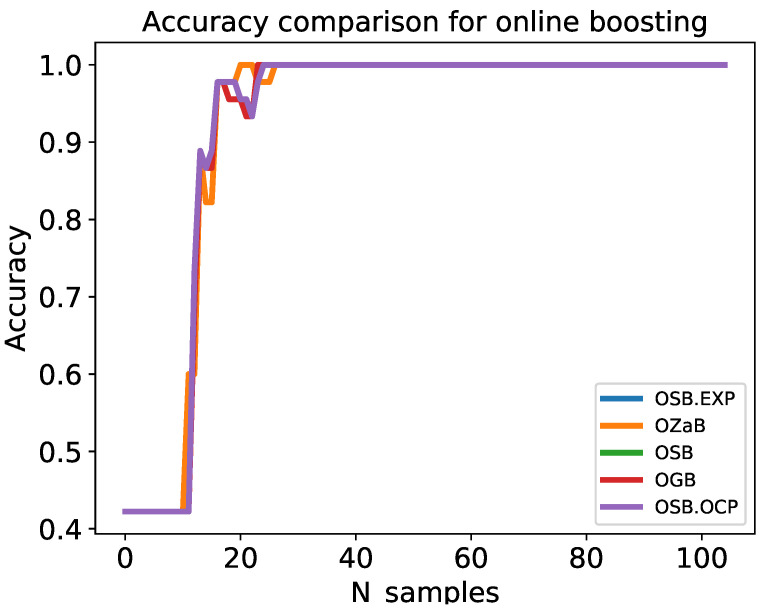
Comparison of online boosting accuracy with an increase in the number of the training samples.

**Figure 9 sensors-22-08422-f009:**
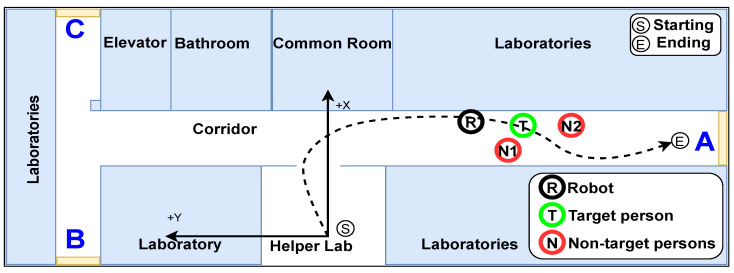
Realistic scenario of the robot and people in the environment.

**Figure 10 sensors-22-08422-f010:**
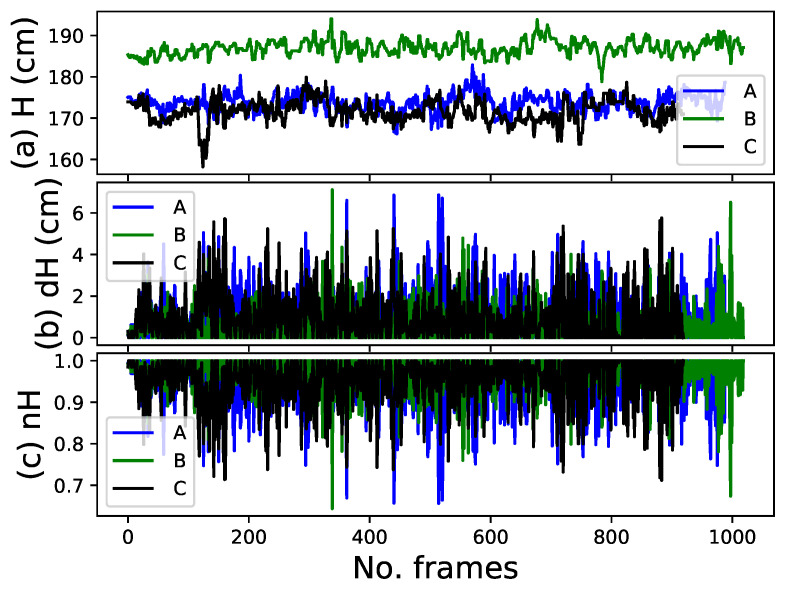
Height feature. (**a**) Height of the target person with respect to the ground plane (**top** plot). (**b**) Height difference of the target person between the current frame and last tracked frame (**middle** plot). (**c**) Normalization of the height difference (**bottom** plot).

**Figure 11 sensors-22-08422-f011:**
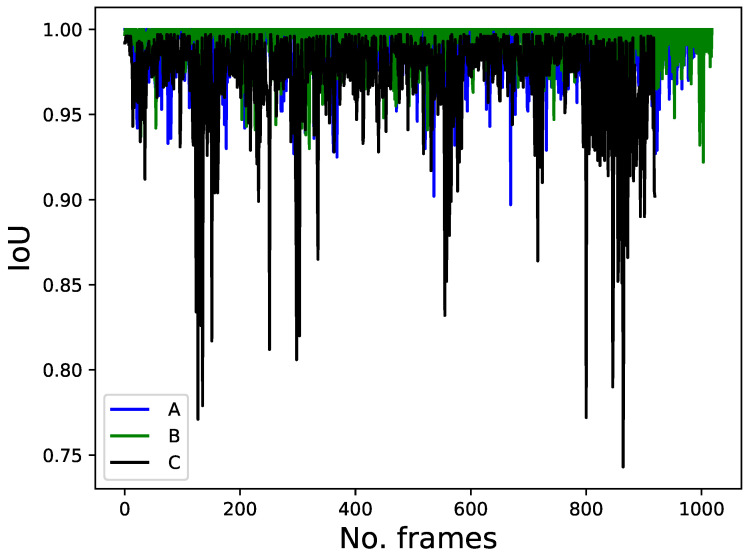
IoU feature.

**Figure 12 sensors-22-08422-f012:**
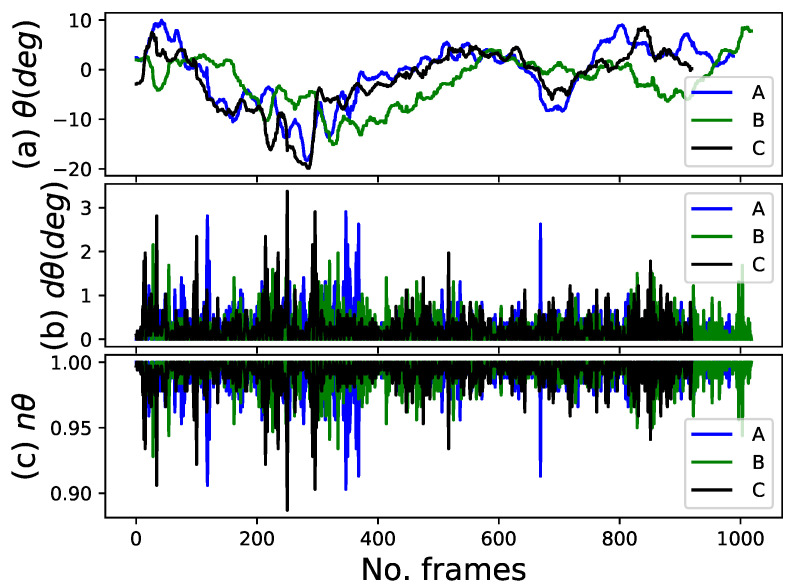
Localization feature. (**a**) Horizontal angle of the target person with respect to the center of the image (**top** plot). (**b**) Angle difference of the target person between the current frame and last tracked frame (**middle** plot). (**c**) Normalization of the angle difference (**bottom** plot).

**Figure 13 sensors-22-08422-f013:**
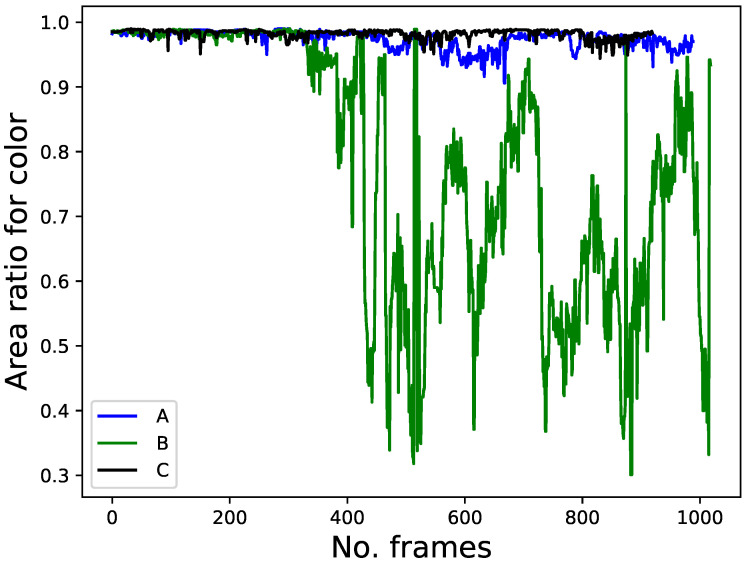
Color feature.

**Figure 14 sensors-22-08422-f014:**
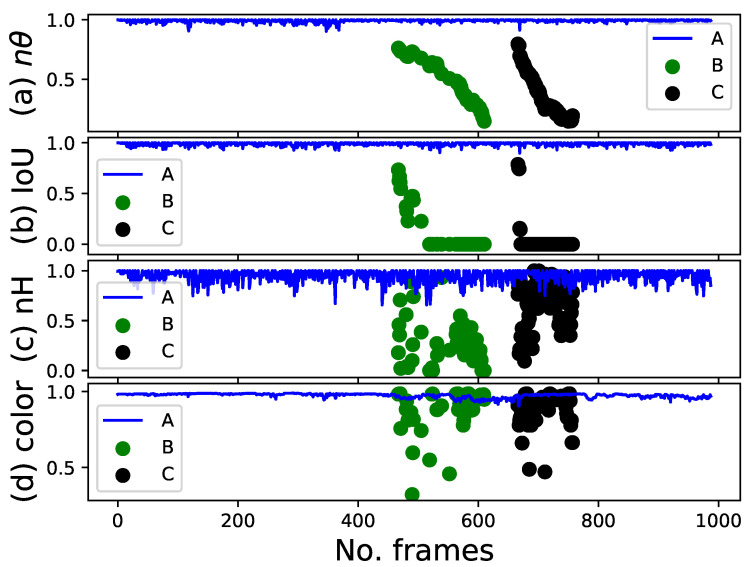
Normalization features for the target person and other people: (**a**) Localization feature (**top** plot). (**b**) IoU feature (**top middle** plot). (**c**) Height feature (**bottom middle** plot). (**d**) Color feature (**bottom** plot).

**Figure 15 sensors-22-08422-f015:**
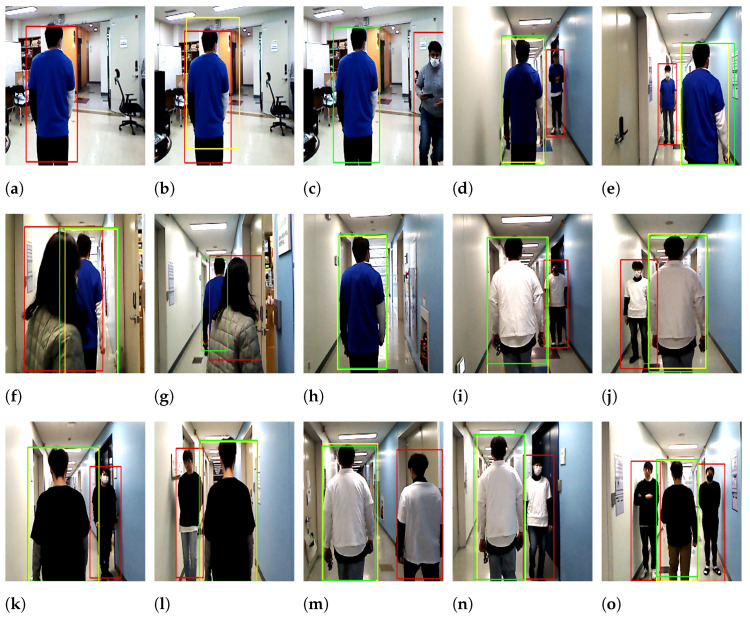
Snapshots of the experiments for three target persons with three different colors: robot’s view.

**Figure 16 sensors-22-08422-f016:**
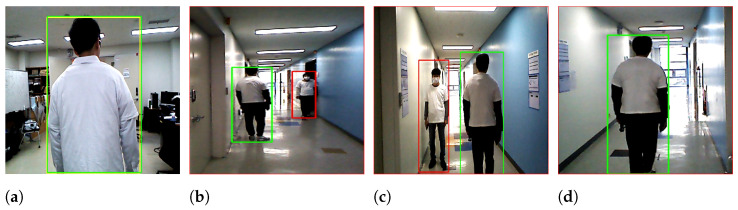
Snapshots of the experiments for target identification under a different lighting environment.

**Figure 17 sensors-22-08422-f017:**
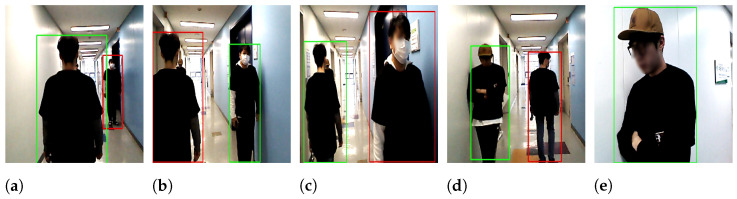
Snapshots of the following failure using only two features of the color and height: robot’s view.

**Table 1 sensors-22-08422-t001:** Quantification of Experimental Results.

No. of Features	Four			Two
Feature name	Color, height, localization and IoU			Color and height
T-shirt color	Blue	White	Black	Black
Successful experiments	13/13	12/13	11/13	8/13

**Table 2 sensors-22-08422-t002:** Experimental results for the blue color.

Parameters	Exp. 1	Exp. 2	Exp. 3	Exp. 4	Exp. 5	Exp. 6	Exp. 7	Exp. 8	Exp. 9	Exp. 10	Exp. 11	Exp. 12	Exp. 13	Total	Average	Std
Experiment status	O	O	O	O	O	O	O	O	O	O	O	O	O	13 /13	-	-
Target’s travel distance (m)	30.40	31.30	31.16	31.39	31.46	31.55	31.40	31.43	31.53	31.44	31.76	31.31	30.61	406.75	31.29	0.38
Robot’s travel distance (m)	29.44	29.52	29.03	29.34	29.37	29.64	29.59	29.26	29.13	29.47	29.64	29.14	26.95	379.51	29.19	0.70
Robot’s travel time (s)	39.93	40.19	40.11	39.18	39.77	39.94	41.41	40.14	38.67	39.21	39.16	38.07	38.28	514.05	39.54	0.91
Robot’s average velocity (m/s)	0.74	0.73	0.72	0.75	0.74	0.74	0.71	0.73	0.75	0.75	0.76	0.77	0.70	-	0.74	0.02
No. of frames (N)	1004	1012	1008	990	995	1003	997	1010	910	969	983	957	959	12,797	984.38	29.12
No. of frames for model ($N_{m}$)	1000	1012	1001	990	980	1000	993	1006	910	969	973	957	932	12,723	978.69	30.40
Successfully tracked (frames) (n)	998	1012	1001	990	979	1000	993	1006	909	969	973	953	859	12,642	972.46	43.76
No. of lost frames by model	2	0	0	0	1	0	0	0	1	0	0	4	73	81	6.23	20.10
Lost frames due to noise	4	0	7	0	15	3	4	4	0	0	10	0	27	74	5.69	7.85
Lost track of the target (frames)	6	0	7	0	16	3	4	4	1	0	10	4	100	155	11.92	26.85
Successfully tracked (s)	39.69	40.19	39.83	39.18	39.13	39.82	41.24	39.98	38.62	39.21	38.76	37.91	34.29	507.86	39.07	1.66
Lost track of the target (s)	0.24	0.00	0.28	0.00	0.64	0.12	0.17	0.16	0.04	0.00	0.40	0.16	3.99	6.19	0.48	1.07
successful tracking rate (%)	**0.99**	**1.00**	**0.99**	**1.00**	**0.98**	**1.00**	**1.00**	**1.00**	**1.00**	**1.00**	**0.99**	**1.00**	**0.90**	**-**	**0.99**	0.03
Lost tracking rate (%)	0.01	0.00	0.01	0.00	0.02	0.00	0.00	0.00	0.00	0.00	0.01	0.00	0.10	-	0.01	0.03
Successful tracking rate for model (%)	**1.00**	**1.00**	**1.00**	**1.00**	**1.00**	**1.00**	**1.00**	**1.00**	**1.00**	**1.00**	**1.00**	**1.00**	**0.92**	**-**	**0.99**	0.02
Lost tracking rate for model (%)	0.00	0.00	0.00	0.00	0.00	0.00	0.00	0.00	0.00	0.00	0.00	0.00	0.08	-	0.01	0.02
fps	25.14	25.18	25.13	25.27	25.02	25.11	24.08	25.16	23.53	24.71	25.10	25.14	25.05	-	24.89	0.51

**Table 3 sensors-22-08422-t003:** Experimental results for the white color.

Parameters	Exp. 1	Exp. 2	Exp. 3	Exp. 4	Exp. 5	Exp. 6	Exp. 7	Exp. 8	Exp. 9	Exp. 10	Exp. 11	Exp. 12	Exp. 13	Total	Average	Std
Experiment status	O	O	O	X	O	O	O	O	O	O	O	O	O	12 /13	-	-
Target’s travel distance (m)	29.54	29.63	29.57	28.26	29.60	29.89	30.16	30.04	30.06	30.12	29.41	29.80	29.35	385.43	29.65	0.54
Robot’s travel distance (m)	27.80	27.95	27.94	26.83	27.83	27.77	27.98	27.70	27.61	27.77	27.43	27.98	27.88	360.46	27.73	0.33
Robot’s travel time (s)	51.12	46.26	45.37	43.64	41.50	47.04	41.88	39.63	37.33	39.21	39.95	39.14	47.36	559.42	43.03	4.12
Robot’s average velocity (m/s)	0.54	0.60	0.62	0.61	0.67	0.59	0.67	0.70	0.74	0.71	0.69	0.71	0.59	-	0.65	0.06
No. of frames (N)	1299	1161	1018	1069	1047	1178	1066	997	943	923	1009	981	1183	13,874	1067.23	110.50
No. of frames for model ($N_{m}$)	1247	1159	1012	1058	1044	1172	1059	990	934	920	994	975	1153	13,717	1055.15	102.06
Successfully tracked (frames) (n)	1212	1159	1010	1032	1044	1172	1059	987	934	918	982	975	1130	13,614	1047.23	97.14
No. of lost frames by model	35	0	2	26	0	0	0	3	0	2	12	0	23	103.00	7.92	12.17
Lost frames due to noise	52	2	6	11	3	6	7	7	9	3	15	6	30	157.00	12.08	14.11
Lost track of the target (frames)	87	2	8	37	3	6	7	10	9	5	27	6	53	260.00	20.00	25.22
Successfully tracked (s)	47.69	46.18	45.01	42.13	41.38	46.80	41.60	39.23	36.97	39.00	38.88	38.90	45.24	549.01	42.23	3.65
Lost track of the target (s)	3.42	0.08	0.36	1.51	0.12	0.24	0.27	0.40	0.36	0.21	1.07	0.24	2.12	10.40	0.80	0.99
Successful tracking rate (%)	**0.93**	**1.00**	**0.99**	**-**	**1.00**	**0.99**	**0.99**	**0.99**	**0.99**	**0.99**	**0.97**	**0.99**	**0.96**	**-**	**0.98**	0.02
Lost tracking rate (%)	0.07	0.00	0.01	-	0.00	0.01	0.01	0.01	0.01	0.01	0.03	0.01	0.04	-	0.02	0.02
Successful tracking rate for model (%)	**0.97**	**1.00**	**1.00**	**-**	**1.00**	**1.00**	**1.00**	**1.00**	**1.00**	**1.00**	**0.99**	**1.00**	**0.98**	**-**	**0.99**	0.01
Lost tracking rate for model (%)	0.03	0.00	0.00	-	0.00	0.00	0.00	0.00	0.00	0.00	0.01	0.00	0.02	-	0.01	0.01
fps	25.41	25.10	22.44	24.49	25.23	25.04	25.45	25.16	25.26	23.54	25.25	25.06	24.98	-	24.80	0.95

**Table 4 sensors-22-08422-t004:** Experimental results for the black color.

Parameters	Exp. 1	Exp. 2	Exp. 3	Exp. 4	Exp. 5	Exp. 6	Exp. 7	Exp. 8	Exp. 9	Exp. 10	Exp. 11	Exp. 12	Exp. 13	Total	Average	Std
Experiment status	O	O	O	O	O	O	X	O	O	O	O	X	O	11 /13	-	-
Target’s travel distance (m)	29.73	29.31	29.80	28.93	28.89	28.83	20.06	29.62	29.25	29.26	29.72	20.44	28.88	362.70	27.90	3.41
Robot’s travel distance (m)	27.78	27.37	26.18	27.51	27.55	27.38	19.22	27.27	27.06	27.17	27.79	19.64	27.49	339.40	26.11	2.99
Robot’s travel time (s)	37.20	39.55	38.90	39.06	40.63	43.12	38.18	42.21	39.14	38.47	41.23	33.10	40.41	511.21	39.32	2.50
Robot’s average velocity (m/s)	0.75	0.69	0.67	0.70	0.68	0.64	0.50	0.65	0.69	0.71	0.67	0.59	0.68	-	0.66	0.06
No. of frames (N)	900	967	951	947	1013	1064	937	977	904	938	977	799	973	12,347	950	62.57
No. of frames for model ($N_{m}$)	890	898	896	898	965	994	859	909	867	901	923	763	942	11,705	900	55.60
Successfully tracked (frames) (n)	873	881	861	875	953	954	839	877	857	886	921	745	938	11,460	882	55.33
No. of lost frames by model	17	17	35	23	12	40	20	32	10	15	2	18	4	245	19	11.37
Lost frames due to noise	10	69	55	49	48	70	78	68	37	37	54	36	31	642	49	19.16
Lost track of the target (frames)	27	86	90	72	60	110	98	100	47	52	56	54	35	887	68	26.43
Successfully tracked (s)	36.08	36.04	35.22	36.09	38.22	38.66	34.19	37.89	37.10	36.34	38.87	30.86	38.96	474.53	36.50	2.25
Lost track of the target (s)	1.12	3.52	3.68	2.97	2.41	4.46	3.99	4.32	2.03	2.13	2.36	2.24	1.45	36.69	2.82	1.09
Successful tracking rate (%)	**0.97**	**0.91**	**0.91**	**0.92**	**0.94**	**0.90**	**-**	**0.90**	**0.95**	**0.94**	**0.94**	**-**	**0.96**	**-**	**0.93**	0.03
Lost tracking rate (%)	0.03	0.09	0.09	0.08	0.06	0.10	-	0.10	0.05	0.06	0.06	-	0.04	-	0.07	0.03
Successful tracking rate for model (%)	**0.98**	**0.98**	**0.96**	**0.97**	**0.99**	**0.96**	**-**	**0.96**	**0.99**	**0.98**	**1.00**	**-**	**1.00**	**-**	**0.98**	0.01
Lost tracking rate for model (%)	0.02	0.02	0.04	0.03	0.01	0.04	-	0.04	0.01	0.02	0.00	-	0.00	-	0.02	0.01
fps	24.20	24.45	24.44	24.24	24.93	24.67	24.54	23.15	23.10	24.38	23.70	24.14	24.08	-	24.16	0.55

**Table 5 sensors-22-08422-t005:** Experimental results using only two features for the black color.

Parameters	Exp. 1	Exp. 2	Exp. 3	Exp. 4	Exp. 5	Exp. 6	Exp. 7	Exp. 8	Exp. 9	Exp. 10	Exp. 11	Exp. 12	Exp. 13	Total	Average	Std
Experiment status	O	X	O	O	X	O	O	O	X	X	X	O	O	8 /13	-	-
Target’s travel distance (m)	29.81	20.15	29.38	29.30	19.85	28.99	29.49	30.07	24.17	24.59	20.25	28.83	28.67	343.54	26.43	4.06
Robot’s travel distance (m)	27.77	19.20	27.39	27.54	19.03	27.26	27.74	28.00	23.41	23.69	19.25	27.95	27.82	326.03	25.08	3.70
Robot’s travel time (s)	41.06	46.16	38.77	41.12	62.76	38.14	38.17	39.44	38.26	34.74	49.97	45.45	49.79	563.83	43.37	7.50
Robot’s average velocity (m/s)	0.68	0.42	0.71	0.67	0.30	0.71	0.73	0.71	0.61	0.68	0.39	0.61	0.56	-	0.60	0.14
No. of frames (N)	983	1173	976	1040	1502	936	961	978	892	843	1262	1126	1253	13,925	1071.15	184.73
No. of frames for model ($N_{m}$)	941	818	913	997	1181	906	935	944	863	838	1044	1049	1144	12,573	967.15	111.51
Successfully tracked (frames) (n)	888	656	908	921	639	905	883	888	765	782	756	991	1059	11,041	849.31	123.09
No. of lost frames by model	53	162	5	76	542	1	52	56	98	56	288	58	85	1532	117.85	147.25
Lost frames due to noise	42	355	63	43	321	30	26	34	29	5	218	77	109	1352	104.00	117.25
Lost track of the target (frames)	95	517	68	119	863	31	78	90	127	61	506	135	194	2884	221.85	249.49
Successfully tracked (s)	37.09	25.81	36.07	36.41	26.70	36.87	35.07	35.81	32.81	32.23	29.93	40.00	42.08	446.90	34.38	4.77
Lost track of the target (s)	3.97	20.35	2.70	4.70	36.06	1.26	3.10	3.63	5.45	2.51	20.04	5.45	7.71	116.92	8.99	10.24
Successful tracking rate (%)	**0.90**	**-**	**0.93**	**0.89**	**-**	**0.97**	**0.92**	**0.91**	**-**	**-**	**-**	**0.88**	**0.85**	**-**	**0.90**	0.04
Lost tracking rate (%)	0.10	-	0.07	0.11	-	0.03	0.08	0.09	-	-	-	0.12	0.15	-	0.10	0.04
Successful tracking rate for model (%)	**0.94**	**-**	**0.99**	**0.92**	**-**	**1.00**	**0.94**	**0.94**	**-**	**-**	**-**	**0.94**	**0.93**	**-**	**0.95**	0.03
Lost tracking rate for model (%)	0.06	-	0.01	0.08	-	0.00	0.06	0.06	-	-	-	0.06	0.07	-	0.05	0.03
fps	23.94	25.41	25.17	25.29	23.93	24.54	25.17	24.80	23.32	24.27	25.26	24.77	25.16	-	24.70	0.65

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
