# Peer review of "Online Boosting-Based Target Identification among Similar Appearance for Person-Following Robots"

_sensors, 2022, doi:10.3390/s22218422_

Round 1

Reviewer 1 Report

The paper proposed a boosting based target identification for person which has important application value.

Although the paper does not have many innovative algorithms, it still has great application potential.

  • It is suggested to add tests under different lighting environments to verify the performance of the algorithm.

Author Response

Response to Reviewer 1 Comments

Dear Reviewer:

We wish to thank you for all your valuable comments in this review. Your comments provided valuable insights, which definitely improved this work. In this document, we try to address the issues raised as best as possible.

Point 1: Although the paper does not have many innovative algorithms, it still has great application potential.

Response 1:  We thank the reviewer for pointing out the novelty of the paper. In this study, the motivation behind current research in person-following is when the target person and other people have similar appearances. Difficult to track the target person among many people who have similar appearances. The proposed system overcomes this challenge by joining multiple fast and efficient features (colors, height, IoU, localization) by online boosting to track the target. Also, we conducted many experiments for people who wore three different T-shirts to evaluate the performance of the system in addition to extra experiments for comparison with previous approaches.

We thank the reviewer for raising this point. We fully agree with the reviewer that the human following robot still has great application potential.

Point 2: It is suggested to add tests under different lighting environments to verify the performance of the algorithm. 

Response 1:  Thank you for reviewer's valuable comment. As the reviewer suggested, we conducted extra experiments to show the capability of target identification under a different lighting environment. These experiments are tested in the early morning when the sunlight passes into the corridor while it does not pass into the lab at all.  So, added Figure 16 to show the capability of target identification under a different lighting environment (Figure 16 and lines (510 - 518).

Author Response

Response to Reviewer 2 Comments

Dear Reviewer:

We wish to thank you for all your valuable comments in this review. Your comments provided valuable insights, which definitely improved this work. In this document, we try to address the issues raised as best as possible.

Point 1: What are the advantages of the online enhancement method proposed in this paper compared with the traditional method? 

Response 2:  We thank the reviewer for pointing out the advantages of the online enhancement method proposed; we added the advantages of the online enhancement method proposed in this paper compared with the traditional method (lines (226 - 230).

Point 2: It is mentioned in this article that “the system cannot effectively identify targets when the same T-shirt having similar height”. If this problem can be solved, please describe the process of solving this problem in details.

Response 2:  We thank the reviewer for raising this point. When the target person and non-target person have the same or similar T-shirts and heights, for example, target A, his height = 175 cm, and target C, his height = 173 cm. The height difference is 2 cm. When we used only two features (color and height), the system cannot effectively identify the target person (see table 5 and figure 17). But when we used four features (colors, height, IoU, localization), the system effectively identify the target person (see tables 2,3,4 and figure 15). 

Point 3: Is the multi-functional framework proposed in this paper also applicable to public databases? Please explain the results of using other datasets.

Response 3:   Thank you for reviewer's valuable comment. We tried our best to design the proposed system so that be generalized and applied to any target person without prior knowledge or pre-training except the deep learning model. The deep learning model dataset is available to the public here (https://github.com/chuanqi305/MobileNet-SSD). So, we have no own databases to be public. Based on your comment, we update our contribution to this paper (lines (62 - 65). 

Point 4: What is the feature extraction method in this paper? Are the feature extracted using the character recognition model representative?

Response 4:  The features used were extracted in this work based on a feature perspective. Also, we added a reference, which explains the feature extraction methods (lines (164 - 165).

Point 5: In the third section, in the process of the threshold selection, have you tried other thresholds? What is the effect?

Response 5:  We thank the reviewer for raising this point. Yes, we have tried other thresholds. For example, the illumination in the lab is approximately uniform, it worked well with the color threshold of 0.5 but it did not work well in the corridor. So, we set up those thresholds empirically for our system and environmental condition after conducting many experiments and trying to set up them well before the final test. Based on your comment,  we explained the effect of thresholds (lines (304 - 311).

Point 6: The format of references is not uniform, it is recommended to unify the format of references.

Response 6: We used your template latex for MDPI that is adopted its own referencing style, titles, and so on.

Reviewer 3 Report

This paper considers a framework for a person identification model that identifies

4 a target person by merging multiple features into a single joint feature online.  However, the contribution is rather trivial and  

and has less theoretical depth.   The novelty is unclear. What is the difficulty of the paper and how to overcome? The comparison with the existing

results should be discussed. The important and recent references are short of. The results are

trivial. Besides, English grammar, spelling,

and sentence structure are too poor to be understood. The tenses in the same paragraph are not unified. The surname and first name in part 

references are wrong.  The uppercase and lowercase letters in the titles of references are chaos.  The fonts of functions and variables are wrongly used.

 The following related nonlinear system tracking control and surface vessels adaptive control and image processing references should be cited to 

highlight the motivation. 

[1] Low-complexity tracking control of strict-feedback systems with unknown control directions, IEEE Transactions on Automatic Control, 2019, 64(12): 5175-5182.

[2] Image enhancement based on rough set and fractional order differentiator, Fractal Fract, 6(4):214, 2020.

[3] Singularity-free continuous adaptive control of uncertain underactuated surface vessels with prescribed performance, IEEE Transactions on Systems, Man and Cybernetics: 

Systems, DOI: 10.1109/TSMC.2021.3129798.

Author Response

Response to Reviewer 3 Comments

Dear Reviewer:

We wish to thank you for all your valuable comments in this review. Your comments provided valuable insights, which definitely improved this work. In this document, we try to address the issues raised as best as possible.

Point 1: However, the contribution is rather trivial and has less theoretical depth.   The novelty is unclear. 

Response 1:  We thank the reviewer for pointing out the novelty of the paper. In this study, the motivation behind current research in person-following is when the target person and other people have similar appearances. 

Point 2: What is the difficulty of the paper and how to overcome it? The comparison with the existing results should be discussed. 

Response 2:  Thank you for the reviewer's valuable comment. Difficult to track the target person among many people who have similar appearances. The proposed system overcomes this challenge by joining multiple features (colors, height, IoU, localization) by online boosting to track the target. For the comparison with the existing work, we explained more in detail in Subsection 4.3.2, entitled “Comparison with Previous Approaches” in the manuscript. 

Point 3: The important and recent references are short of. 

Response 3:  As the reviewer suggested, we added important and recent references. 

Point 4: The results are trivial. 

Response 4:  We fully agree with the reviewer that our experimental results did not present well. So, we added a link to show you the results clearly (line 354). 

Point 5: Besides, English grammar, spelling, and sentence structure are too poor to be understood. The tenses in the same paragraph are not unified. 

Response 5:  As the reviewer raised issues regarding the manuscript's English, we put our manuscript to the English editing service by a native speaker and get an official certificate.

Point 6: The surname and first name in part references are wrong. The uppercase and lowercase letters in the titles of references are chaos.  The fonts of functions and variables are wrongly used.

Response 6: We thank the reviewer for raising this point. We used your template latex for MDPI that is adopted its own referencing style, titles, and so on.

Point 7: The following related nonlinear system tracking control and surface vessels adaptive control and image processing references should be cited to 

highlight the motivation.

[1] Low-complexity tracking control of strict-feedback systems with unknown control directions, IEEE Transactions on Automatic Control, 2019, 64(12): 5175-5182.

[2] Image enhancement based on rough set and fractional order differentiator, Fractal Fract, 6(4):214, 2020.

[3] Singularity-free continuous adaptive control of uncertain underactuated surface vessels with prescribed performance, IEEE Transactions on Systems, Man and Cybernetics: Systems, DOI: 10.1109/TSMC.2021.3129798.

Response 7:  As the reviewer suggested three papers, we added those papers as references [6-9].  

Reviewer 4 Report

The paper needs to improve better in terms of organization.  The experiments and quantitative results presented were not clear. What do you suggest? The results section must contain quantitative results in the form of a table and these must be discussed in detail to convey to the reader how these results are obtained and why they are valuable. 

The manuscript needs to include a description of the metrics used in their experiments, their mathematical formula, etc. Ex: accuracy, precision, recall, and f-measure. Or, include a separate table to show these metrics. Also the Images, Figures, Tables, Captions, etc. formatted are not organized.- The Conclusion section needs to be more comprehensive.

Author Response

Response to Reviewer 4 Comments

Dear Reviewer:

We wish to thank you for all your valuable comments in this review. Your comments provided valuable insights, which definitely improved this work. In this document, we try to address the issues raised as best as possible.

Point 1: The paper needs to improve better in terms of organization. 

Response 1:   We thank the reviewer for raising this point. We reorganized the paper where we moved the online boosting algorithms evaluation from section 3 to section 4. Also, we rewrote the subtitle.

Point 2:   The experiments and quantitative results presented were not clear. What do you suggest? 

The results section must contain quantitative results in the form of a table and these must be discussed in detail to convey to the reader how these results are obtained and why they are valuable. 

Response 2:  We fully agree with the reviewer that the experiments and quantitative results presented were not clear. So,  we added Table 1 to show the quantitative results.

Point 3: The manuscript needs to include a description of the metrics used in their experiments, their mathematical formula, etc. Ex: accuracy, precision, recall, and f-measure. Or, include a separate table to show these metrics. 

Response 3:  As the reviewer suggested, we added extra metrics (precision, recall, and f-measure)  to evaluate the online boosting model (lines (356 - 357) and equations that show how to evaluate our system (lines (191 - 192).

Point 4: Also the Images, Figures, Tables, Captions, etc. formatted are not organized.

Response 4:  We thank the reviewer for raising this point. We used your template latex for MDPI that is adopted its own referencing style, titles, and so on.

Point 5: The Conclusion section needs to be more comprehensive.

Response 5:  As the reviewer suggested, we added some sentences in the conclusion section to be more comprehensive.

Round 2

Reviewer 2 Report

Agreed to accept after being modified and meeting the publishing requirements

Author Response

Response to Reviewer 2 Comments (Round 2)

Dear Reviewer:

We wish to thank you for all your time and valuable comments on this review. Your comments provided valuable insights, which definitely improved this work.

Reviewer: Agreed to accept after being modified and meeting the publishing requirements.

Authors: Thank you so much for your agreement to accept our responses.

Reviewer 3 Report

This authors revised the manuscript according to my comments. However, the contribution
is still weak. The writing is poor. The grammar of main texts is questinable. The  format of and size of function and variables are wrong. The surname and name of the authors in [18] are incorrect. The commas shold be followed by a space. In (10), min(A_i, B_tl) is
min(|A_i|, |B_tl|)? The uppercase and lowercase letters of the title in References are choas. 

Author Response

Response to Reviewer 3 Comments (Round 2)

Dear Reviewer:

We wish to thank you for all your time and valuable comments on this review. Your comments provided valuable insights, which definitely improved this work. In this document, we try to address the issues raised as best as possible.

Point 1: The authors revised the manuscript according to my comments. However, the contribution is still weak.

Response 1:  Thank you for the reviewer's valuable comment and time. To the best of our knowledge, our method is the first approach that identifies the target person using color, height, location, and IoU features and merges those features into a single feature online using the OzaBoost algorithm to solve similar appearance cases for the person-following robots.  

Point 2:  The writing is poor. The grammar of main texts is questinable.

Response 2:   As the reviewer raised issues regarding the manuscript's English, our manuscript was reviewed by a native speaker at the first as well as we put our manuscript to the English editing service by a native speaker to get an official certificate, you can find the official certificate on the first page of the PDF of this revised version.

Point 3:  The  format of and size of function and variables are wrong. The surname and name of the authors in [18] are incorrect. The commas shold be followed by a space. The uppercase and lowercase letters of the title in References are choas. 

Response 3:  Thank you for the reviewer's valuable comment and time.  We used your template latex for MDPI to write our manuscript. So, we cannot change the format and size of functions and variables or the uppercase and lowercase letters of the title and so on in References.

Point 4:  In (10), min(A_i, B_tl) is min(|A_i|, |B_tl|)?

Response 4:  Thank you for the reviewer's valuable comment and time. In (10), it is a min(A_i, B_i)  because the areas are positive in our case.